# The Known, the Unknown and the Future of the Pathophysiology of Endometriosis

**DOI:** 10.3390/ijms25115815

**Published:** 2024-05-27

**Authors:** Maria Ariadna Ochoa Bernal, Asgerally T. Fazleabas

**Affiliations:** 1Department of Obstetrics, Gynecology & Reproductive Biology, Michigan State University, Grand Rapids, MI 49503, USA; ochoaber@msu.edu; 2Department of Animal Science, Michigan State University, East Lansing, MI 48824, USA

**Keywords:** endometriosis, origins, pathophysiology, microRNA, cytokines, fibrosis, infertility, baboon model, mouse model

## Abstract

Endometriosis is one of the most common causes of chronic pelvic pain and infertility, affecting 10% of women of reproductive age. A delay of up to 9 years is estimated between the onset of symptoms and the diagnosis of endometriosis. Endometriosis is currently defined as the presence of endometrial epithelial and stromal cells at ectopic sites; however, advances in research on endometriosis have some authors believing that endometriosis should be re-defined as “a fibrotic condition in which endometrial stroma and epithelium can be identified”. There are several theories on the etiology of the disease, but the origin of endometriosis remains unclear. This review addresses the role of microRNAs (miRNAs), which are naturally occurring post-transcriptional regulatory molecules, in endometriotic lesion development, the inflammatory environment within the peritoneal cavity, including the role that cytokines play during the development of the disease, and how animal models have helped in our understanding of the pathology of this enigmatic disease.

## 1. Introduction

Endometriosis is an inflammatory, fibrotic and estrogen-dependent gynecological disorder characterized by endometrial-like tissue outside the uterus, mainly in the pelvic peritoneum, ovaries, and rectovaginal septum. In rare cases, this endometrial-like tissue can be found in other areas, such as the pericardium, pleura, diaphragm or even the brain [1,2,3,4]. It affects approximately 5–10% of women of reproductive age, and it is associated with pelvic pain and infertility [2,5]. Endometriosis is a common condition with a profound negative impact on the lives of women during their fertile years. Over their lifetime, women with this disorder confront challenges that affect their quality of life, participation in daily activities, relationships, work productivity, well-being, and mental health [6]. Endometriosis is usually classified according to the criteria formulated by the American Society of Reproductive Medicine (ASRM). Their criteria include four stages: minimal stage (stage I), mild (stage II), moderate (stage III) and severe (stage IV). These stages include the lesion size, location and extent of the tissue growth from the minimal to the severe stage [7,8]. However, this staging system does not correlate the pain symptoms and the risk of infertility [8,9].

Endometriosis is an underdiagnosed disease that is associated with a delay from the onset of symptoms that can take up to 7–8 years before diagnosis [10], and the symptoms can vary widely. Women may be asymptomatic or present with a single symptom or a combination of symptoms with different intensities that can easily be attributed to other conditions [5]. Some of the symptoms that are associated with endometriosis include painful menstruation (dysmenorrhea), cyclical or non-cyclical abdominal pain, recurrent painful urination (dysuria), pain during and after sexual intercourse (dyspareunia), painful defecation (dyschezia), gastrointestinal discomfort [11] and decreased libido [12,13]. Currently, there are no reliable biomarkers available to diagnose this disease. The gold standard for the diagnosis of endometriosis has been surgical assessment by laparoscopic visualization [5]. Current endometriosis guidelines are challenging laparoscopy as the gold standard and imaging modalities are becoming more important in clinical practice. Laparoscopy would be now recommended in patients with negative imaging results and/or where empirical treatment was unsuccessful or inappropriate [14].

## 2. Postulated Origins

The etiology of endometriosis is complex and there are several contributing factors leading to the development of this disorder. There are numerous theories that have been put forward to explain the origin of endometriosis (Figure 1).

### 2.1. Sampson’s Theory: Retrograde Menstruation

Retrograde menstruation is a process whereby menstrual blood flows backward via the fallopian tubes into the pelvic cavity, resulting in peritoneal seeding of menstrual tissue [15]. Sampson’s retrograde menstruation theory is the most widely accepted cause of endometriosis. In 1927, Sampson hypothesized that endometriosis is the result of the reflux of endometrial fragments through the fallopian tubes during menstruation, with subsequent attachment and implantation of endometrial fragments, which results in peritoneal and ovarian lesions [16]. Retrograde menstruation occurs in up to 90% of women within their reproductive lifespan [17], but only 6% to 10% of women develop endometriosis [1,18].

Although retrograde menstruation happens in most women of reproductive age, the immune system is usually able to clear the endometrial fragments, preventing their growth in ectopic sites. However, when the immune system fails to clear the fragments, endometriosis occurs [19]. One possible hypothesis is that endometriosis can be caused by the decreased clearance of the endometrial cells and fragments due to reduced natural killer (NK) activity, and/or decreased macrophage activity [20,21].

There are numerous factors that need further study in terms of the development of endometriosis, such as a dysregulation of the clearance mechanisms, altered immune function or other exogenous factors.

### 2.2. Coelomic Metaplasia

This theory was initially introduced by Iwanoff and Meyer [22,23]. The coelomic metaplasia theory proposes the ability of normal cells, which are derivatives of the primitive peritoneum, to differentiate into endometrial tissue. This theory is based on the fact that the pelvic peritoneum and ovaries are derived from the epithelium of the coelomic wall [23]. This theory is used to explain endometriosis in females with the absence of menstruation, rare cases of endometriosis in males who undergo estrogen therapy for prostate cancer, adolescent girls, premenopausal women, or women with total abdominal hysterectomies [22,23,24,25]. This theory would also explain the occurrence of endometriosis in patients with Mayer–Rokitansky–Küster–Hauser (MRKH), a syndrome that is associated with the lack of a uterus [26,27].

### 2.3. Embryonic Rest Theory

This theory, proposed by Von Recklinghausen and Russell in the 1890s, suggests that embryonic cell rests of Mullerian origin within the peritoneal cavity could differentiate into functioning endometrial tissue under the appropriate stimuli [22,28]. This theory also explains the presence of rare cases of endometriosis in men, since the embryonic cell rests of Mullerian origin are also present in males [29] or in any location along the migration pathway of the Mullerian system.

### 2.4. Vascular and Lymphatic Metastasis

In accordance with Sampson’s theory, the endometrial tissue is usually spread through the fallopian tubes by retrograde menstruation [16]. However, this theory does not explain implants of endometriotic tissue outside the peritoneal cavity. The lymphatic dissemination theory has been proposed to explain the observations of endometrial tissue moving through the vasculature and lymph nodes [30]. Lymphatic system involvement may explain the reason why endometrial tissue can be found in rare areas outside the peritoneal cavity or ovaries, such as the pleura or the pericardium [31]. In this case, understanding the role of the lymphatic system in endometriosis is essential to establish novel therapeutic approaches for the disease.

### 2.5. Tissue Injury and Repair Theory (TIAR)

TIAR theory postulates that endometriosis is caused by trauma. In this context, TIAR represents an estrogen-related mechanism that is pathologically magnified in a reproductive organ that is already estrogen-sensitive [32]. The trauma could be caused by chronic uterine peristaltic activity, or hyperperistalsis, a mechanical trauma resulting in increased desquamation of tissue fragments of the basal endometrium [33], which could induce micro-traumatization at the endometrium–myometrium interface, activating TIAR and causing an increase in local production of estrogen. Estrogen, in addition, could act as positive feedback, causing the self-perpetuation of the disease, allowing the endometrial tissue to implant outside the uterine cavity [34].

### 2.6. Quinn’s “Denervation–Reinnervation” Theory

The denervation–reinnervation theory proposes that endometriotic cells can be found outside the uterine cavity because of uterine nerve injuries and uterosacral ligaments. These may be caused by difficult intrapartum episodes and continued strain during defecation [35]. The damage to the uterine nerves causes denervation, leading to a loss of fundo-cervical polarity and uterotubal dysmotility. The ectopic endometrial cells from retrograde menstruation adhere to the damaged tissue in the peritoneal cavity and the uterosacral ligaments. The posterior tissue repair, including the reinnervation of the uterine isthmus, vaginal and uterosacral ligaments, promotes chronic pelvic pain, dysmenorrhea, and subfertility, sometime after the primary injuries [36].

### 2.7. Stem Cell Theory

Stem cells are undifferentiated cells that can self-renew and generate into more differentiated cells. These types of cells are usually present in the endometrial tissue [37]. Endometrial stem cells may travel to an ectopic location via different paths: retrograde menstruation from the shedding endometrium, lymphatic and vascular dissemination, direct migration and invasion or a combination of both [22]. The stem cell theory postulates that the endometrial stem cells residing in the basalis layer of the endometrium [38], at menstruation, can reflux through the fallopian tubes and lymphatic and/or vascular dissemination, establishing endometriotic lesions outside the peritoneal cavity [22,38]. On the other hand, there are some studies that suggest that the bone marrow-derived stem cells may play a role in the development of the disease [15,39].

### 2.8. Genetic/Epigenetic Theory

The causes of endometriosis remain largely unknown, but heritability plays a role in the genetic component. It is estimated that heritability can contribute up to 50%, with 26% due to common genetic variation [40]. The genetic/epigenetic theory is a more recent theory that contributes to the endometriotic disease theory (EDT) [40]. The EDT states that genetic events are required for the development of endometriosis. The genetic/epigenetic theory adds to this and suggests that the genetic and epigenetic changes and the redundancy of cellular processes promote changes that contribute to the disease [41]. The redundancy of many different cellular mechanisms that can be regulated by multiple signaling pathways can explain the cumulative effect of sequential genetic and epigenetic incidents [42]. Other studies have also shown that acetylation and methylation may also play an important role in the modulation of the genotype in endometriotic tissue [43,44,45,46]. There are two main key histone modifications responsible for differential gene expression that can alter transcriptional activity. These alterations in the gene expression could have a direct effect on cell cycle growth, cell cycle arrest and apoptosis, and they may play an important role in the pathogenesis of endometriosis [47].

## 3. Pathophysiology of Endometriosis

### 3.1. Hormonal Dysregulation

In most mammals, endometrial function is primarily governed by both steroid hormones, estrogen and progesterone. Both hormones play an important role in regulating the menstrual cycle. An imbalance in both ligands and/or their receptors can cause several gynecological disorders, including endometriosis [48]. As mentioned at the beginning, endometriosis is an estrogen-dependent gynecological disorder characterized by endometrial-like tissue outside the uterus. Modifications mediated by estrogen play an important role in the etiology of endometriosis. Women with endometriosis often present with estrogen dominance caused by a local estrogens synthesis and an increase of estrogen receptor activity in endometriotic cells [49]. Although the origin of the disease is not clear, there is an obvious hormonal dysregulation associated with this disease [4].

#### 3.1.1. Estrogen Dominance

The presence of estrogen plays a key role in endometriosis. Estrogen enhances the survival or persistence of the endometriotic tissue outside the uterine cavity. The four different types of natural estrogens (E2) are estradiol, estrone, estriol and estetrol. There are three major sources in the body that can produce estrogen in women with this disease (Figure 2). Estradiol is secreted by the ovaries, which can reach the attached endometrial fragments through the circulation. Aromatase in adipose tissue catalyzes the conversion of circulating androstenedione to estrone and subsequently converts it to estradiol, which can reach endometrial implants. Endometriotic tissue can also express a complete set of steroidogenic genes, including aromatase, allowing local conversion of cholesterol to estradiol [50,51]. Aromatase, which regulates the conversion to estradiol, produces an increase in the local estrogen concentration, enhancing the growth of endometriotic tissue and maintenance of endometriosis.

Estrogen binds to the estrogen receptor (ESR), allowing its entrance into the cell. Estrogen acts via both ESR1 and ESR2. Both are expressed in human endometrial stromal and epithelial cells. Although both are present in the endometrium, ESR1 seems to be a more prominent mediator of the estrogenic action in this tissue, whereas ESR2 is expressed at a lower level [52,53]. Since the effects of estrogen are primarily manifested through these receptors, their expression levels are critical in the assessment of estrogen action in endometriosis.

Some studies suggest that ESR1 and ESR2 could act in different ways to promote the proliferation of endometrial cells and tissue-invasion activity to promote ectopic lesions. ESR1 might be involved in the initiation of endometriosis, while the overproduction of estradiol in endometriosis would drive ESR2 signaling to support endometriotic tissue survival and enhance inflammation [54]. Several studies have reported lower levels of ESR1 and higher levels of ESR2 in human endometriotic tissues and primary stromal cells compared with endometrial tissue and cells [55].

Studies in the baboon model of endometriosis have shown a time-dependent reduction in ESR1 within eutopic endometrial stromal cells a month after the induction of endometriosis. This decrease was statistically significant within six months of the disease and remained low throughout the time course of the disease until termination at 15 months. In the case of ESR2, this was reduced in both epithelial and stromal cells. These decreased levels of ESR1 and ESR2 could contribute to a non-receptive uterine environment, which could affect embryo implantation [56].

#### 3.1.2. Progesterone Resistance

Progesterone (P4) is a steroid hormone that is produced by the adrenal cortex and gonads, with the corpus luteum being the major source of P4 during the menstrual cycle. P4 not only plays a key role in the female reproductive tract but also plays a vital role in the maintenance of the uterus during pregnancy [57]. Progesterone signaling is implicated in follicular growth, ovulation and luteinization. It is also responsible for the differentiation of endometrial epithelial cells, resulting in the development of the secretory endometrium and the differentiation of endometrial stromal cells into the decidual phenotype to prepare the uterus for embryo implantation [58]. Progesterone signaling is compromised in women with endometriosis [59,60,61]. This abnormal P4 signaling in the endometrium plays a significant role in endometrial receptivity by interfering with the regulation of the uterine epithelial proliferation and impairing decidualization [62,63]. Another molecular consequence of the P4 resistance associated with endometriosis would be the increased invasiveness of endometriotic tissue to promote the establishment of the ectopic endometrial implants [63,64].

Genetic and epigenetic studies have suggested that the eutopic endometrium of women with endometriosis responds differently to circulating P4 compared to the endometrium of women without the disease [65,66]. This was also observed and confirmed in the baboon model of endometriosis [61,67,68].

The term “Progesterone resistance” is due to the failure of endometrial tissue to properly respond to progesterone signaling [66]. Most of the actions of P4 are mediated through the binding and activation of progesterone receptors (PGRs). These progesterone receptors, PGR-A and PGR-B, are members of the superfamily of ligand-activated transcription factors. Several studies have shown that the levels of both receptors, particularly PGR-B, are significantly lower in the endometrial lesions in women with endometriosis compared with the eutopic endometrium [69,70]. In addition, there is direct evidence to support the fact that microRNA dysregulation [71,72] and the promoter hypermethylation of PGR-B [73,74] are potential mechanisms for the loss of PGR-B in women with endometriosis.

### 3.2. Inflammatory Response and Immune Dysregulation in Endometriosis

Endometriosis is defined as an inflammatory condition. The immune system is believed to play an important role in the pathophysiology of endometriosis [75]. An altered proinflammatory immune environment in the endometrium in women with endometriosis and a lack of proper immune surveillance in the peritoneum may support the attachment and progression of endometrial tissue, promoting the development of this disease [1,58]. Endometriosis is suggested to be associated with dysfunction or suppression of different populations of immune cells. Most of them have been studied in the peritoneal cavity [76] and eutopic endometrium of women with endometriosis [77].

The next section is focused on the immune abnormalities that have been observed in patients with endometriosis. The most relevant cells, cytokines and factors involved in the progression of endometriosis are discussed below.

#### 3.2.1. Proinflammatory Environment: Immune Cells

The immune endometrial environment is extremely dynamic. The morphology of the endometrium, proliferation, and differentiation of the cellular components during the menstrual cycle stimulate the traffic of different immune cell populations [58]. Moreover, it has to be considered that not all the endometrial immune populations vary consistently during the menstrual cycle in women with or without endometriosis [77].

The leukocytes (CD45+) are distributed in the reproductive tract in either an aggregated or a dispersed form in the epithelial layer, lamina propria and stroma [78]. They represent a subpopulation of between 6 and 20% in the female reproductive tract [79]. Under normal conditions, peritoneal leukocytes remove ectopic endometrial cells from the peritoneal cavity. However, with endometriosis, the ability to eliminate these cells is significantly decreased, promoting implantation, proliferation and the recruitment and activation of peritoneal macrophages [80]. This situation facilitates an environment that may promote the progression and development of endometriotic lesions [81].

Most of the immune cells in the endometrium are cells that are resident in the tissue. Some of them are derived from the peripheral circulation. The predominant immune cells are T cells, macrophages, dendritic cells (DCs), natural killer (NK) cells, neutrophils and mast cells [78]. These cells play an important role during endometrial remodeling and repair and facilitate the clearance of endometrial cells and tissue after menstrual shedding [82]. Some of these cells may not succeed in their task and allow endometrial fragments to implant at ectopic locations via retrograde flow and facilitate the establishment of endometriosis [81].

T cells

T cells make up around 1–2% of the total endometrial cells [58]. T cells are one of the most important contributors to endometriosis [83]. They are part of the adaptative immune system, affecting the activity of other immune cells through a variety of cytokines secreted by these cells [58,84].

Different subtypes of T cells are involved in the immune response to endometriosis through different mechanisms [83]. Depending on the surface marker and function, different types of T cells have been identified. The most relevant in endometriosis are T helper cells (Th) and regulatory T cells (Treg). Both express CD4 and each of them is characterized by the type of cytokine that they produce. Some of the subcategories that are included in the T helpers are Th1, Th2 and Th17 [85].

Th1 cells stimulate the production of IL-2, IFN-γ, lymphotoxin, and other cytokines that promote the action of macrophages and natural killer [86] cells. There is a lack of knowledge regarding Th1 in the endometrium of women with endometriosis. In peripheral blood, Th1 cells are more abundant in women with endometriosis when compared with women without the disease [87]. The presence of these cells is associated with the severity of the disease; more specifically, with the secretion of IL-2 and IFN-γ, which have been found to be considerably elevated in women with deep infiltrating endometriosis [86].

Th2 stimulates the production of IL-4, IL-5, IL-6, IL-10 and IL-13. IL-4 and IL-10, which are cytokines that are known to collaborate with B cells, trigger humoral immune responses, which recruit and activate other cells to sites of inflammation [86]. IL-4 and IL-13 are also cytokines involved in the differentiation of resident fibroblasts into myofibroblasts. This suggests that Th2 cells may be involved in the process of fibrosis [88], which is one of the hallmarks of endometriosis.

Th17 are cells derived from naïve CD4+ lymphocytes in the presence of TGF-β and IL-6 [88]. These T helpers are characterized by the production of cytokines such as IL-17, IL-17A, IL-21, and IL-22 [89]. They are present in blood and the peritoneal fluid at different stages of endometriosis. The percentage of Th17 cells in the peritoneal fluid in women with advanced stages of endometriosis has been shown to be higher than in women with mild endometriosis [90]. They appear to contribute to the disease severity and promote the proinflammatory environment characteristic of endometriosis [58]. Some studies have shown that the elevated levels of IL-17A, in the plasma and peritoneal fluid of women with endometriosis could play an important role in the progression of endometriosis, stimulating the production of other cytokines that are involved in the process of angiogenesis and chronic inflammation [91,92].

Regulatory T cells (Treg) are an important population of cells that promote the proinflammatory environment in women with endometriosis. They can regulate the Th1/Th2 response [93], macrophages, mast cells and natural killer cells, amongst other types of immune cells [77]. They are present in ectopic lesions and in the peritoneal fluid of women with endometriosis, and they appear to be increased in with women who suffer from endometriosis compared with women without the disease. This has also been observed in the baboon model [94]. Some studies have shown that Treg cells in the peritoneal fluid have a positive correlation with the stage of the disease [95].

Macrophages

Macrophages play an important role in the immune response. They are capable of stimulating the synthesis and secretion of a variety of molecules that can alter the function of neighboring cells [81]. Activated macrophages can secrete IL-1, IL-6, IL-8, IL-10, IL-12 and IL-13, together with other cytokines [58]. The two main phenotypes to which macrophages can polarize to are the classical phenotype (M1) and the alternate phenotype (M2) [96]. M1 macrophages are known to secrete cytokines that promote the initial response against infections and trigger the inflammatory response. On the other hand, M2 macrophages manifest as anti-inflammatory and tissue-specific, primarily promoted by IL-4, IL-10, IL-13 and transforming growth factor-β (TGF-β). Studies have shown that M2 macrophages are involved in tissue repair during inflammation, mainly in the peritoneal environment of women with endometriosis [97,98,99].

When considering that the macrophage phenotypes in women with endometriosis compared to women without endometriosis are different, and given the complexity of the nature of the endometriotic lesion environment, it is most likely that the macrophage M1–M2 polarization in endometriosis may depend on a variety of functional states in the disease [96].

Dendritic cells (DCs)

DCs are antigen-presenting cells essential for the initiation and maintenance of the T cells’ immune response [81]. They produce cytokines such as IL-6, IL-10, and RANTES, among others, that have an important role in the proinflammatory environment [100], especially in endometriosis. There are studies showing that activated DCs are significantly higher in women with stage I–II endometriosis compared with women without the disease, which suggests that these cells are critical players in the development of the immune response [101]. DCs can play a role in endometriotic lesions by increasing neuroangiogenesis and contributing to the lesion growth and pain in women with this disease. In general, immature DCs are higher in the eutopic endometrium in women without endometriosis, suggesting a potential failure of DC maturation in women with endometriosis, which could lead to a failure in the clearance of the endometrial cells shed during menstruation, promoting the progression of the disease [58].

Uterine natural killer (uNK) cells

uNK are the predominant leukocyte population in the normal endometrium. They play an important role, not only in endometriosis but also in infertility [102]. Different studies have shown that changes in uNK cells in the blood, peritoneum and endometrium could play a role in the development of endometriosis [103]. These cells, in the normal endometrium, have low cytotoxic activity. However, in women with endometriosis, their activity is even more reduced than in the normal endometrium. In cases where a woman with endometriosis is also infertile or presents with recurrent pregnancy loss, the uNK cytotoxic activity is increased. This suggests that uNK cells play an important role during the maintenance and the establishment of pregnancy and could serve as an indicator of endometriosis-related infertility and recurrent miscarriage [58].

#### 3.2.2. Cytokines and Growth Factors

The cascade of events that involve the inflammatory response is an important aspect of the development of endometriosis. As previously mentioned, there is clear evidence that endometriosis is related to an abnormal function of different immune cells that produce specific cytokines. Cytokines and growth factors are proteins produced by immune cells that are usually increased in the peritoneal fluid of women with endometriosis, which contributes to the pathogenesis of the disease [104]. Some of the cytokines that seem to play a crucial role in the establishment and lesion survival in endometriosis are described below.

Interleukin-6 (IL-6)

IL-6 is one of the most studied proinflammatory cytokines in endometriosis and is one that is highly expressed in patients with the disease [105,106,107]. It is primarily produced during acute and chronic inflammation. IL-6 is increased in the endometrium of women with endometriosis and in ectopic lesions [77,108]. It affects the secretion of other cytokines, promotes the activation of T-lymphocytes and promotes the proliferation of B-lymphocytes [109]. Macrophages can secrete IL-6 in response to different substances in the peritoneal fluid. Endometrial epithelial and stromal cells produce IL-6 depending on the hormonal and immunological environment [110]. The increased levels of IL-6 in the peripheral blood of patients with endometriosis also suggest that this could be an important cytokine that could be used as a serum marker for non-surgical prediction of endometriosis [111].

Interleukin-8 (IL-8)

IL-8 is an activator of neutrophils and a very potent angiogenic cytokine [112]. There are several studies that have shown an increased concentration of IL-8 in the peritoneal fluid and the serum of women with endometriosis when compared to women without the disease. It stimulates the adhesion of endometrial cells to fibronectin, implying a role for it in the pathophysiology of the disease by promoting cell attachment and proliferation. It has also been observed that the elevated IL-8 levels in women with endometriosis might be correlated with the severity of the disease [113].

Interleukin-1 (IL-1)

IL-1 is mainly produced by monocytes and macrophages and plays an important role during the regulation of the inflammation and immune responses. The IL-1 family consists of two different molecular forms, IL-1α and IL-1β [114]. IL-1 is a pleiotropic cytokine that may promote the development of endometriosis in the peritoneal cavity, upregulating the expression of other cytokines such as IL-6 or other growth factors. Also, it has been shown that IL-1α is a potent stimulator of matrix metalloproteinase enzymes (MMPs) during the proliferative phase in the endometrium [115]. IL-1 increases the secretion of MMP-1, MMP-2, MMP-3, and MMP-9 in uterine endometrial cells from women with endometriosis [114]. The IL-1 family also includes cytokines such as IL-33 [116]. Studies have suggested that women with endometriosis, specifically the ones who suffer from deep infiltrating disease, have high levels of IL-33 in their peritoneal fluid and serum compared with women without the disease [117]. In addition to this, other studies have shown that human endometriotic lesions produce IL-33 and that the levels of IL-33 are significantly elevated in the endometriotic tissue of advanced-stage patients compared to healthy, fertile controls [118].

Tumor necrosis factor alpha (TNF-α)

TNF-α is a pleiotropic cytokine that plays an important role during the inflammatory process involved in the progression of endometriosis. It is mainly produced by macrophages, uterine natural killer cells, and Th1 cells, together with other cells [77]. In the human endometrium, TNF-α is implicated in the physiological process of endometrial proliferation and shedding [119]. It has also been observed that it can activate inflammatory leukocytes and stimulate macrophages to produce other cytokines, such as IL-1 and IL-6, which further enhances the proinflammatory environment in women with endometriosis. In addition to this, it induces the expression of other enzymes, such as COX2, which can stimulate cell proliferation and angiogenesis, promoting endometrial tissue growth [77]. Studies have reported the presence of significantly higher levels of TNF-α in patients with endometriosis at an early stage of the disease [120]. TNF-α may be one of the essential players in the pathogenesis of endometriosis development.

Transforming growth factor-β (TGF-β)

TGF-β is an inflammatory growth factor that plays an important role during the development of endometriosis. Different studies have shown an increase in this growth factor in the peritoneal cavity of patients with endometriosis when compared to women without the disease, implying a role during the pathogenesis of the disease [121,122]. TGF-β is one of the most widely studied molecules in fibrosis. There are three different isoforms (TGF-β1, TGF-β2 and TGF-β3). Even though there are a variety of cell types that respond to and produce TGF-β, TGF-β1 is the one that has been mainly associated with the development of fibrosis [88]. TGF-β is a very powerful chemoattractant growth factor for macrophages, leukocytes and fibroblasts [109]. Some studies have reported that the increasing levels of TGF-β in the peritoneal cavity of women with endometriosis could be associated with the increased survival, invasion and proliferation of ectopic endometrial cells during endometriotic lesion development [123].

### 3.3. The Fibrotic Component during Endometriosis

During retrograde menstruation, the process of the menstrual tissue traveling into the pelvic cavity can result in peritoneal and ovarian disease, followed by inflammatory responses, scarring and fibrosis [124]. Fibrosis is defined by the development of fibrous connective tissue resulting from continuous tissue damage and repair.

Myofibroblasts are crucial to the process of fibrosis. Once these cells are activated, they proliferate and produce a collagenous extracellular matrix that heals but at the same time, disrupts and modifies the surrounding structures. This process is defined surgically as scarring [125]. The activation of myofibroblasts in endometriotic lesions can be detected based on the presence of α-smooth muscle actin (α-SMA). During the development of fibrosis, the lesions can lose the characteristic features of endometrial glands and stroma, complicating the histologic confirmation of the disease.

There are two factors that seem to be critical for the activation of myofibroblasts: transforming growth factor and the stiffness of the tissue [126]. When the myofibroblasts are activated, they can induce an increase in proliferation, migratory ability, production of cytokines and interstitial matrix that stimulates changes in the environment. Continuous myofibroblast activity causes the accumulation and contraction of the collagenous extracellular matrix, enhancing fibrosis. This process will eventually result in a hypertrophic scar, causing the disruption of the anatomy of the tissue [127].

#### Fibrosis in Endometriotic Lesions

The peritoneal cavity is one of the most common areas where endometrial lesions can be established. They contain an abundant amount of smooth muscle that seems to represent an important characteristic of peritoneal endometriotic lesions. As was previously mentioned, the TGF-β levels are significantly increased in the peritoneal fluid of women with endometriosis when compared to women without the disease. One of the events that has been observed in mesothelial cells is that when they are exposed to TGF-β, the production of lactate is increased, which produces a reduction in the pH of the environment. This increase in the production of lactate promotes the acidic activation of the TGF-β ligand, causing a secondary induction of myofibroblast differentiation [128].

A different type of lesion is the ovarian cyst. It is known that fibrosis can be present in the ovarian cyst walls. The pseudo-capsule of the cyst is primarily constituted of fibrotic tissue. Fibrosis can also be identified in the ovarian cortex surrounding the endometrioma. Some studies have also shown that cells derived from ovarian endometriosis-activated platelets can promote epithelial to mesenchymal transition (EMT), fibroblast to myofibroblast transdifferentiation and differentiation to smooth muscle cells. This results in an increase in the cell contractibility, collagen production and fibrosis via TGF-β/Smad signaling [129].

### 3.4. Infertility

Infertility is defined as the inability of a couple to conceive after one year of unprotected intercourse. Infertility is one of the primary concerns related to endometriosis. In healthy couples, the probability of achieving pregnancy in any single month is around 15–20%. In women with undiagnosed or untreated endometriosis, this probability is closer to 2–10% [109]. It is estimated that 35–50% women with this disease are infertile [1] and that 25–30% of all infertile women have endometriotic lesions as the only identifiable cause of their infertility [130]. Women with endometriosis have a higher risk of infertility and miscarriage, due in part to endometrial abnormalities [131]. The distorted anatomy that scarring and the local inflammation can cause could also lead to infertility [58].

The distortion due to the scarring and fibrosis, in combination with the inflammatory response, promote a hostile environment in the pelvic cavity and uterus that contributes to infertility in patients with endometriosis by affecting oocytes, sperm, and embryos. In addition to this, women with this disease who achieve pregnancy have worse pregnancy outcomes than women without endometriosis. This could be a consequence of the proinflammatory environment previously described in the endometrium and the effects during the processes of nidation and placentation [132]. Other possible causes of infertility in women who have endometriosis could be adhesions, disturbed folliculogenesis, luteal phase defects, progesterone resistance and anti-endometrial antibodies [133,134,135]. Despite the clear association between endometriosis and infertility, the mechanisms associated with the cause of infertility remain unclear.

In the literature, there are several tools, including the endometriosis fertility index (EFI), that help predict pregnancy rates in patients with surgically documented endometriosis who attempt non-IVF conception. This tool is only helpful for those infertility patients who have had surgical staging of their disease, but it is not designed to predict any aspect of endometriosis-associated pain [136]. One factor found to predict pregnancy that is not included in the EFI is the uterine abnormality. This is because it is uncommon to find it in infertile patients with endometriosis. However, if this condition is found, it is not taken into consideration [136]. The EFI provides a score from zero to ten and the score predicts the results of subsequent non-IVF treatments [136]. It is a simple, robust and validated clinical tool that is very useful in developing treatment plans for infertile patients with endometriosis.

There are several studies that support the association between endometriosis and infertility even in early stages of the disease. Nearly 50% of women with minimal or mild endometriosis will be able to achieve pregnancy without any treatment. However, only 25% of women with moderate endometriosis will conceive spontaneously, while only a few spontaneous conceptions will occur in severe cases of endometriosis [137]. Superficial peritoneal lesions are more closely related to infertility than cases of deep infiltrating endometriosis [133]. Studies in animal models have also been able to replicate the association between endometriosis and subfertility. Studies that include the rabbit as an animal model have shown that injecting peritoneal fluid from animals affected by endometriosis into normal animals results in a decrease in their implantation sites, implying that substances in the peritoneal cavity of the animals affected by endometriosis could impact fertility [138]. The same effect was observed in fertile mice when they were injected with human peritoneal fluid from infertile women with endometriosis [139]. In addition to this, studies using the mouse model suggest that the development of endometriosis can cause implantation failure and defective decidualization, as occurs in humans, impacting fertility [140]. Furthermore, data from the baboon model suggest that in the induced model of endometriosis, an increased angiogenic capacity, decreased apoptotic potential, progesterone resistance, estrogen hyper-responsiveness, and an inability to respond appropriately to embryonic signals contribute to the reduced fecundity associated with this disease [56].

As previously mentioned, peritoneal factors play an important role during the development of endometriosis. The inflammatory environment associated with macrophages and the production of cytokines can affect processes such as ovulation, embryo quality and implantation [109]. As described earlier, IL-6 plays an important role in endometriosis. It is a cytokine that is normally low during the proliferative phase and high during the implantation window. Some studies have shown that the expression of IL-6 was reduced in a group of women experiencing unexplained recurrent miscarriages when compared to fertile women [141].

Another aspect of endometriosis to consider is the P4 resistance that occurs during the development of the disease. P4 and E2 are essential for the establishment of a successful pregnancy. The P4 resistance and dysregulation of hormonal signaling during the development of endometriosis could be another contributing factor that impacts implantation by altering the endometrium and affecting fertility [63]. Embryo implantation is a rigorously controlled process that is regulated by E2 and P4 in the majority of mammals [142]. A balance between E2 and P4 is required to achieve a successful implantation. The progesterone resistance that occurs in endometriosis could interfere with the expression of progesterone-induced proteins critical for implantation and endometrial receptivity, which could contribute to infertility related to endometriosis [143].

TFG-β is a growth factor that is crucial for female reproduction. A dysregulation of the TFG-β signaling pathway may have catastrophic consequences, leading to reproductive diseases, disturbing embryo implantation, ovulation or decidualization [144]. There is evidence that TFG-β may play a role in the etiology of endometriosis. TFG-β signaling is associated with angiogenesis, EMT and cell cycle control, all of which are significantly increased in the peritoneal cavity of women with endometriosis compared to those women without the disease [145]. TFG-β is also known as a mediator of P4 action during the secretory phase [146], and it modulates the expression of critical proteins, including leukemia inhibitor factor (LIF), implying a mechanism by which fertility might be compromised [109].

It is known that fertility is compromised in some women with endometriosis, even at the early stages, while other women have minimal effects. As previously mentioned, the balance between E2 and P4 is crucial for a normal implantation and the loss of this equilibrium during endometriosis could be one of the causes of infertility. Management of infertility caused by endometriosis will be discussed later in this review.

## 4. Role of MicroRNAs in the Pathophysiology of Endometriosis

Different studies have suggested that miRNAs play a role in non-malignant and malignant diseases that involve the female reproductive tract. Abnormal expression of different miRNAs has been observed in several reproductive tract diseases, including preeclampsia, endometrioid endometrial adenocarcinoma and endometriosis [147,148].

### 4.1. Biogenesis of miRNAs

miRNAs are small, noncoding regulatory RNAs that are not translated into proteins [149]. MicroRNAs contain ~20 nucleotides that can regulate gene expression and play a fundamental regulatory role in several pathological processes [150,151,152]. In 1993, Rosalind Lee, Rhonda Feinbaum and Victor Ambros presented the first evidence of what are now known as miRNAs. Ambros’s group described a 22-nucleotide RNA encoded by lin-4, a gene in *Caenorhabditis elegans* involved in larval development that does not code for a protein but can instead bind to the lin-14 transcript and regulate its expression [150,152,153]. In the canonical miRNA biogenesis pathway (Figure 3), genes are transcribed by RNA polymerase II or RNA polymerase III to produce a primary microRNA transcript (pri-miRNA). This transcript, while still in the nucleus, is processed into a smaller precursor miRNA (pre-miRNA) by the microprocessor complex Drosha-DGR8. The resulting precursor is translocated from the nucleus to the cytoplasm by Exportin-5-RAN-GTP. Once there, the Dicer-TRBP complex cleaves the pre-miRNA into a mature single-stranded miRNA. The mature miRNA binds to its mRNA target at their complementary sequence to reduce the expression of the target protein by inhibiting mRNA translation to proteins or simply by decreasing the mRNA levels [149,154,155].

The study of miRNAs is crucial to the understanding of the pathophysiology of different diseases. Over the years, several studies have reported that the altered expression of miRNAs is associated with cancer, cardiovascular diseases, diabetes, fibrosis and gynecological diseases such as endometriosis [156].

### 4.2. Relevant miRNAs in Endometriosis

Studies have supported the hypothesis that miRNAs play an important role during the progression of endometriosis [147]. Some miRNAs, such as miR-29c, -451, -141, -21, and -200a, have been shown to be altered in baboons with endometriosis [61,157]. These studies were also validated in women with endometriosis [157]. miRNAs can be involved in processes such as progesterone resistance, inflammation, cell proliferation, extracellular matrix remodeling and angiogenesis, amongst other processes that are key to the development of endometriosis [109].

Some miRNAs have been shown to be associated with infertility. MiR-29c has been demonstrated to contribute to progesterone resistance. This miRNA is upregulated in endometriosis, resulting in the decreased expression of one of its targets, the FK506-binding protein 4 (FKBP4) gene. FKBP4 is a co-chaperone that optimizes the function of the progesterone receptor. The upregulation of miR-29c, resulting in the decrease of FKBP4 during the window of implantation, could lead to progesterone resistance in women with endometriosis, promoting infertility [61].

Another miRNA that has been reported to be altered in endometriosis in baboon studies and in women with endometriosis is miR-451. This miRNA, through its targets, is involved in the suppression of apoptosis and cell proliferation and invasion, which are hallmarks of endometriosis [157]. The downregulation within endometriotic lesions of this miRNA leads to significant increases in the expression of its predicted target, YWHAZ. YWHAZ codes for the protein 14-3-3ζ, known to suppress apoptosis, enhance proliferation and promote invasion, which are hallmarks of endometriosis [157].

Different studies have shown a significant elevation in the serum of miR-451 in women with confirmed endometriosis [158,159]. This shows the potential utility of mirR-451 as a possible serum diagnostic marker for endometriosis diagnosis, although definitive confirmatory studies remain to be performed.

Another major process that has been studied in relation to miRNAs and endometriosis is angiogenesis. MiR-15a-5p was reported to be suppressed in endometriotic lesions and proposed to be involved in angiogenic events associated with endometriosis [160].

Another miRNA that plays a role in endometriosis is miR-210 and its targets IGFBP3 and COLA8A1. This miRNA has been shown to be downregulated in the ectopic endometrium of women and baboons with endometriosis, while IGFBP3 and COL8A1 expression was increased in the ectopic endometrial tissue. MiR-210 suppression was shown to contribute to endometriotic lesion development by increasing cell proliferation and migration [161].

## 5. Management and Treatment in Current Clinical Practice

An accurate classification of endometriosis is crucial to assess the state of the disease. The clinical manifestations of endometriosis are varied and the relationship between the symptoms and the severity of this disease is unclear [162]. There are currently different classification systems available to classify endometriosis. The revised American Society for Reproductive Medicine (rASRM) classification is the most widely used classification of endometriosis [162]. However, it does not consider the involvement of deeply infiltrating endometriosis in different sites. Therefore, in order to include that component, the ENZIAN classification was developed [163]. Both classifications have their strengths and weaknesses, but both of them have a common end point of classifying endometriosis. The use of ENZIAN has been very useful for both surgical approaches and diagnostics and aims to enable documentation of surgical findings, ultrasound, and MRI clearly and objectively [164].

Various studies have described different medical and surgical therapies for endometriosis. Laparoscopy is currently the gold standard for visualizing lesions and detecting endometriosis, but ideally, it should be linked to a confirmed positive histology [165]. The treatment for endometriosis includes two important etiologies: pain relief and amelioration of infertility. In some cases, medical therapies that are used to manage pain, in general, are not useful for managing infertility [166].

When the goal is to enhance fertility, medical treatment is not the best option. They are hormonal and mainly focus on blocking ovulation and estrogen production. When the main goal is managing pain, medical treatment is one of the best options, although surgery might also be a possibility, depending on the severity of the pain in the patient [5].

### 5.1. Management of Pain

Endometriotic lesions are often associated with underlying fibrosis and distortion of the surrounding anatomy, which can lead to pain [167]. The sensitivity to E2 in endometriosis involves the growth of nerve endings into the endometrial lesions, which could also have an influence on the activity of the neurons throughout the central nervous system, leading to pain [168]. Depending on the severity of the pain, medical and/or surgical procedures might need to be implemented.

As endometriosis is an estrogen-dependent disease, medical treatments have focused on establishing a hypo-estrogenic or hyper-progestogenic environment [5]. It is important to mention that medical treatment usually does not eradicate the disease and the chronic pelvic pain associated with endometriosis involves repeated courses of medical therapy, surgical intervention, or both. In most cases, the symptoms reappear after the therapy is discontinued [11].

During management of the pain, the desire of the endometriosis patient to conceive naturally is not usually immediate. In this case, combined oral contraceptives pills containing estrogen and progestin are provided to patients. These contraceptives induce a central inhibition of gonadotropin secretion, inhibiting ovulation and reducing ovarian estrogen secretion. In this scenario, the combined oral contraceptives promote a hyper-progestogenic environment, inducing decidualization and atrophy of the ectopic endometrium [169]. Gonadotropin-releasing hormone (GnRH) agonists can also be administered to suppress hypothalamic and pituitary function, producing a hypo-estrogenic state. They are usually very effective in treating pain [170]. The most widely used agents are GnRH agonists or antagonists and oral contraceptives [1].

The surgical approach to relieving pain in endometriosis can sometimes be used as a first option or initiated following failed medical therapy [2]. Surgical procedures usually include excision, laser ablation of endometriotic implants on the peritoneal surface, excision or drainage or ablation of endometriomas and interruption of nerve pathways [2]. This conservative surgery (no removal of the ovaries or uterus) is usually performed in women who wish to maintain their ability to conceive. The aim of this approach is to eliminate all the visible endometriotic lesions and try to restore the anatomy. Although the effect on the pain after performing this type of surgery is usually satisfactory, symptoms may reoccur after surgery [171]. However, this does not always imply the recurrence of the disease.

For those endometriosis patients who suffer from pain and for whom the desire to conceive is no longer an issue, hysterectomy or a bilateral salpingo-oophorectomy is also an option. Hysterectomy for chronic non-specific pelvic pain associated with endometriosis is an alternative for many women [172]. Some studies have observed a significant long-lasting reduction in pain symptoms after this procedure [173].

In addition to this, there are studies that have reported that about 30% of endometriosis patients suffer from chronic pelvic pain after the excision of the endometriotic lesions in the pelvic area [174,175]. Recent studies focused on adolescents and young adults with endometriosis have identified plasma proteins and biological pathways associated with persistent pelvic pain post-surgical treatment of endometriosis [176]. These new studies could provide new opportunities for improved personalized treatment and faster symptom improvement to manage the pain in young women with endometriosis. They could also provide new targets for treatments.

### 5.2. Management of Infertility

The treatment options for women with endometriosis related to infertility are usually medical treatment, surgery or assisted reproductive techniques. The use of hormonal treatments used for pain management is usually contraindicated due to their contraceptive effects [133]. One main medical strategy in the treatment of endometriosis-associated subfertility to improve pregnancy outcome is the use of ovulation suppression agents, although studies have shown that this type of treatment works well for pain but there is no clear evidence that it contributes to the improvement of fertility [177].

Surgery is another option for infertility management associated with endometriosis. Ablation of endometriotic lesions is recommended for the treatment of infertility related to stage I or II endometriosis [165]. In moderate or severe endometriosis, the goal of the surgery is to restore the normal anatomy of the pelvis and remove large endometriomas [133]. The benefit of medical treatment before or after surgery is not clear, but the suppression of endometriosis before surgery may promote the reduction of inflammation. In women with ovarian endometriomas, undergoing surgery for infertility or pain can increase the spontaneous post-operative pregnancy rate, depending on the methodology used for its management [178].

In any case, the risks and the benefits associated with surgical procedures for endometriosis associated with infertility must be carefully evaluated.

Assisted reproductive techniques (ART) such as IVF can bypass the fallopian tubes, increasing the chances of pregnancy. It is currently the most successful treatment for those cases of infertility associated with endometriosis. These techniques are highly recommended, especially in cases where the tubal function is compromised [179], but the cost of these treatments could be a limitation for some women [180].

## 6. Animal Models to Study the Disease

Limitations related to studying endometriosis in women due to the extended time to diagnosis have made animal models indispensable for studying the early cellular and molecular mechanisms associated with this disease. For years, animal models have allowed us to investigate the mechanisms and regulation of pathways of different diseases in a controlled manner. While there are known disadvantages of extrapolating data from animals to humans, animal models have been useful in studying different events involved in the etiology and pathophysiology of endometriosis [181].

Non-primate animal models of induced endometriosis have been crucial in the study of endometriosis. Some of the animals in this group include rabbits [182,183,184], rats [185,186,187] and mice [188,189]. These models are used to study the mechanisms involved in endometriosis that are not possible to examine in larger mammals. Also, the animal care facilities, handler training, and housing are more cost-effective and manageable as opposed to larger and more expensive animal models.

Several non-human primate models, including macaques [190,191] and baboons [192], have been extremely useful for studying endometriosis. Since they menstruate, they have the advantage of providing a phylogenetically similar animal model to the human. In the case of the baboon model, this animal has an endometrial morphology, physiology, and menstrual cycle similar to women [21]. Baboons can also develop spontaneous endometriosis, which makes them one of the most suitable and appropriate models for studying this disease [192,193].

The next sections are primarily focused on highlighting the advantages of the baboon and mouse models of endometriosis.

### 6.1. Baboon Model

The development of animal models of endometriosis has been, and still is, crucial for the investigation of the disease pathogenesis and the development of new targets for treatments [194]. The mechanisms underlying the development and the maintenance of this disease remain unclear. This lack of understanding, as well as the ethical considerations and limitations associated with human studies, increase the demand for a suitable model to study this disease [109].

Although none of the current animal models have been completely successful in mimicking all the aspects of this disease, they have been proven to be very valuable tools.

The non-human primate model of endometriosis, particularly the baboon, has demonstrated an unquestionable relationship between women with this disease, revealing similar reproductive physiology, immunology and establishment and progression of endometriosis, specifically as it relates to peritoneal lesions, which is the most common form of endometriosis found in women [192,193]. Although the cost of maintenance for these animals is more expensive than most other species, the baboon offers clear advantages for crucial studies focused on endometriosis [192].

Along with this, baboons can spontaneously develop endometriosis resembling the ectopic lesions in humans [195]. These lesions have been found to range from the minimal to the disseminated form in the baboon, similar to the different stages of the disease found in women [192,196]. Another advantage of the use of the baboon as a model is that endometriosis can be induced by injecting autologous menstrual tissue into the pelvic cavity of the same animal. The injection of the menstrual tissue into the intraperitoneal cavity mimics the physiological process of retrograde menstruation, which also allows the study of the progression of the disease from its very initial stages [56,192,195]. In addition to this, the use of the baboon permits multiple and complex surgical procedures and collection of biological samples during the time course of the disease without the need for a hysterectomy.

All these advantages make the baboon an excellent model to study the pathogenesis of endometriosis (Appendix A) [21,56,62,67,68,94,107,129,157,158,192,194,195,197,198,199,200,201,202,203,204,205,206,207,208,209,210,211,212,213,214,215,216,217,218,219,220,221,222,223,224,225,226,227,228,229,230,231,232,233,234,235,236,237,238,239,240,241,242,243,244,245,246,247,248,249,250,251,252,253,254,255,256,257,258,259,260,261,262,263,264,265,266,267,268,269,270,271,272,273,274,275,276,277,278,279,280,281,282,283,284,285,286,287,288,289,290], including the identification of components that may play a role in the establishment of the disease and the development of endometriotic lesions. These advantages can provide a clearer understanding of the mechanisms that contribute to the alteration of the endometriotic environment that could result in pain, fibrosis, or lead to infertility in the context of endometriosis.

This primate also offers an important preclinical model to test drugs for the prevention or treatment of the disease. Studies to evaluate therapeutical effects can alternatively be performed in animals that already have the disease (spontaneous or induced), allowing the performance of both paired (before and after treatment) and unpaired (treatment vs. positive control and negative control) comparisons [193].

Taking into consideration all these advantages, the baboon model has been utilized in a significant number of studies [56,107,157,204,216,217,218,227,228,232,234,240,241,242,248,271,291,292]. Some of these studies include the assessment of the efficacy of statins, such as simvastatin, in the baboon model with endometriosis. The statins represent an important alternative approach for the potential treatment of endometriosis in women for whom hormonal treatments are ineffective or cause undesirable effects [278].

In addition to this, other studies have also used the baboon model of endometriosis to validate aberrant miRNA expression in the pathogenesis of endometriosis [61,157]. This has not only been helpful in validating the altered expression of miRNAs in baboon and human tissue but has also been crucial for those studies that have demonstrated differentially expressed miRNAs levels in the serum of patients with endometriosis as a potential non-invasive diagnostic test [158,279]; however, this needs further validation.

### 6.2. Mouse Models

As mentioned previously, humans and certain primates, such as baboons and macaques, have similar reproductive biology. They have a similar menstrual cycle length as well as cyclical endometrial changes and embryological similarities [293]. Although there are strong advantages to using non-human primate models of endometriosis, the cost can limit the options for several researchers.

Due to this, the mouse model of endometriosis has also been very useful for studying endometriosis [189]. Although they are a non-menstruating species and do not develop endometriosis spontaneously, rodent models have been and still are extremely useful for transgenerational studies to understand the potential impact of endometriosis in women with this disease.

There are immune responses and hormonal regulation that play an important role in the pathogenesis of endometriosis that differ in mice compared to humans, which could present some challenges; however, the mouse model is widely used due to the numerous advantages. One of them is the small size, the short estrous cycle and the length of gestation allowing them to have new offspring once per month. In addition to this, the ease of the genetic manipulation and regulation of specific gene targets makes the mouse a well suited model for dissecting and investigating different pathways of disease pathogenesis [188].

#### Induction Methods for the Development of Endometriosis in Rodents

As previously mentioned, mice do not menstruate. To facilitate the study of endometriosis, the ideal mouse model would be one that presents a mechanism that allows the endometrial tissue to translocate to the peritoneum via retrograde menstruation, as initially proposed by Sampson in 1927 [16]. Currently, there is no such model, but there are others that have been developed over the last several years that facilitate the study of endometriosis.

Suture method

The suture method consists of the suturing or adhesion of uterine tissue onto the peritoneum of a recipient or the same mouse. This endometriosis model allows for the easy location of the lesions and measurement of their sizes. This model also allows the use of autologous, syngeneic, or heterologous engraftments.

Autologous and syngeneic models use murine tissue donors. The uterine tissue engrafted is usually sutured to the peritoneal wall [294,295].

The suture method has advantages but also has several limitations. One of the challenges that it presents is that the surgically implanted lesions completely bypass the attachment phase of the disease. This model does not allow for the study of spontaneous lesion attachment or varying of the location of development of the lesion [188]. In addition, the suture material and the healing process associated with this type of surgery may alter the normal process of lesion development and may alter how angiogenesis develops [296].

Injection method

The injection method involves the injection of uterine fragments through a surgical opening into the peritoneal cavity. This method consists of the introduction of minced tissue taken from one of the uterine horns of the same mouse or a donor mouse. The uterine horn is cleaned by stripping all the fat and muscle. The pieces are usually less than 1.5 mm [297,298].

Following surgery, the recipient mouse can develop multiple lesion types. Typically, lesions will form around the injection site, and they may even form a cesarean scar. Lesions are typically light pink or cystic, but white fibrotic lesions can also be found. The lesions are usually found attached to the peritoneal wall, intestinal mesentery, behind the stomach or spleen and perivesical adipose tissue [110].

One of the advantages of this method is that it allows the study of the initial stages of endometriosis, which includes the process of angiogenesis, defective apoptosis, endometrial proliferation, and the inflammatory environment [299].

Menstruating mouse model

In humans, the functional layer of the endometrium undergoes the process of decidualization during the late secretory phase and ends with endometrial shedding during the process of menstruation [300]. Mice, on the other hand, have estrous cycles and do not menstruate spontaneously (with the exception of the spiny mouse [301]). The mouse endometrium does not spontaneously decidualize and shed. However, there have been studies that have been able to develop a mouse model of menstruation [302] and use it to study the development endometriosis [295].

Published studies have described a mouse model of endometriosis using syngeneic mouse menstrual donor tissue introduced into the peritoneum of an immunocompetent recipient mouse [295,303,304]. One of the advantages of this menstruating mouse model is that the lesions recovered from these mice have very well-organized stromal and glandular structures. In addition to this, this model has demonstrated the importance of the inflammatory microenvironment in the presence of the shed endometrium, which could be useful in understanding the role of inflammation in the development of endometriosis [295].

## 7. Importance of the Study of Endometriosis and Considerations

The present review has principally focused on the origins, the pathophysiology, the role of microRNAs, and animal models related to studying this disease. Despite all the recent research and all the advances related to endometriosis, fundamental issues concerning the pathophysiology and early diagnosis remain unsolved.

The first is to fully understand how endometriosis is initiated. The different theories presented in this review try to explain the origin of endometriosis, but the etiology of endometriosis is complex. There are several factors that contribute to the development of the disease, but the uniqueness of each individual must be considered. There is a considerable delay between the presence of the first symptoms in a woman and the diagnosis of the disease. This is partially caused by the inability of the patient to make a clear distinction between “normal” and “abnormal” menstrual experiences [305] and partially due to the physician’s reliance on diagnostic methods that may not sufficiently address all the patient’s concerns [306]. For a woman, the time spent with no diagnosis could be as harmful as any clinical management related to the chronic pelvic pain without knowing the origin of the pain [305]. The etiology of endometriosis is still unclear, but understanding the origins of the disease is necessary to reduce the time of diagnosis, ensure the efficacy of the treatments and improve the quality of life of the women.

Despite the improvements in the diagnosis and efforts to understand this disease, many women still face ineffective, non-invasive treatments and diagnostic methods that cause multiple side effects and that do not always resolve the pain in patients with endometriosis [307,308]. Therefore, investigating in more detail the contribution of the inflammatory response and the proinflammatory environment could be the key to improving the treatments and diagnostic methods. Endometriosis is defined as an inflammatory condition. In this review, we have described the different immune cells, cytokines and growth factors that could contribute to an altered proinflammatory environment in the endometrium of women with endometriosis. Research has shown that some of these proinflammatory cytokines have been considered as potential non-invasive biomarkers for diagnosis [309,310] or possible targets for new treatments of endometriosis [311,312]. However, some of the findings have been challenging and contradictory due to the heterogeneous nature of the disease [313]. Targeting the inflammatory component of this disease should be one focus of future research. There is still a lack of knowledge regarding the regulation of cytokines from the perspective of disease progression and the role they play in the process. There are no current large-scale studies using cytokine arrays to analyze the dynamic changes in the inflammatory response during the progression of endometriosis [308]. This could certainly help us to understand the role of cytokines in lesion development and cell invasion.

The study of the proinflammatory environment and the immune cells is also crucial not only to further understand the development of the disease but also to identify potential links with other pathologies such as tissue damage and repair. Macrophages can play an important role in tissue repair, regeneration and fibrosis development [314,315]. Endometriosis has recently been defined as a fibrotic condition in which endometrial stroma and epithelium can be identified [3]. The consistent presence of fibrosis in this disease plays a crucial role in the pathogenesis. The development of fibrosis in endometriotic lesions is a complex phenomenon with mechanisms that still remain unclear. In the past several years, scientists and physicians have been more aware of the need to orient the focus toward the factors that contribute to fibrosis. Recent studies using magnetic resonance imaging have shown the potential use of collagen-targeting probes to efficiently detect fibrotic lesions in the mouse model [316], which could be used to inhibit the signaling pathways that promote fibrosis. Cellular and molecular pathways related to the development of fibrosis could be a critical target for treating women with endometriosis. The mechanisms responsible for the onset of fibrosis in different forms could be used to lead to new potential targets for novel therapies.

Fibrosis is consistently present in peritoneal, ovarian and deep infiltrating endometriosis, leading to symptoms of pain and reduced fertility [3]. The causes of infertility in patients who suffer from endometriosis can vary. They can range from anatomical distortions due to the tissue adhesion and fibrosis to endocrine irregularities and immunological disturbances. In the majority of cases, the accumulation of these events seems to interact with mechanisms that are still unclear [133]. Despite extensive research, no consensus agreement has been reached, although several mechanisms have been proposed to explain the association between endometriosis and infertility. Fertility preservation techniques seem to be a valuable option for women undergoing surgical interventions for endometriosis, such an endometrioma excision. This type of surgery has demonstrated a negative impact on the ovarian reserve. Future research should focus on fertility-conserving surgical methods to reduce the diminished ovarian reserve postoperatively [317].

The reality is that there is not an actual cure for endometriosis, only treatment options to manage the symptoms. It is a complex disease that affects many women, regardless of their ethnicity or social status. Enhanced awareness, followed by early diagnosis and management, could help to slow the natural progression of the disease and decrease the burden of its symptoms. There are knowledge gaps that still exist. There is a critical need for non-invasive diagnostic methods as well as medical treatments that alleviate pain and do not interfere with the fertility of the patient.

## 8. Outlook for Research on Endometriosis

Despite years of research, our knowledge of endometriosis and our ability to successfully treat the disease are limited by the incomplete understanding of the pathogenesis of the disease [318]. The diverse clinical and molecular aspects of endometriosis likely due to inconsistencies in disease classification and the variability among control populations remain an enigma. Achieving reliable models to mimic the development of endometriosis is essential to revealing not only molecular and cellular mechanisms underlying this disease but also its etiology.

New approaches in the field of endometriosis, such as RNA-sequencing (RNA-seq) [319], transcriptome analysis [320], or single-cell RNA-sequencing (scRNA-seq) [321,322,323,324], could help to characterize this heterogeneity among the population and facilitate the assessment. RNA-seq has been increasingly considered a very useful analytical technique within the field of endometriosis. The amount of data produced provides the opportunity to analyze gene expression or even aspects related to the biological roles of small RNAs, including miRNAs [325]. Different studies have used the “-omics” technologies to investigate the gene expression profile of the genome of the eutopic endometrium from women with endometriosis [101,326,327]. Some of these studies have been able to show the different cell subtypes present within the eutopic endometrium microenvironment, especially the immune cell profiles among the different stages of endometriosis [101]. Single-cell analysis platforms have been very useful for identifying cell populations within the endometrium. This might be critical for more effective therapies for endometriosis [328]. Single-cell RNA-sequencing, single-cell ATAC-sequencing and spatial transcriptomics have become extremely important for studying endometrial cells [329] or the spatial dynamics of the human endometrium [330].

Recently, spheroids and organoids have been providing a newer technology to study endometriosis. These models have been shown to be excellent models for dissecting different mechanisms associated with endometriotic lesion development and to provide experimental investigations such as genetic manipulation. Moreover, 3D models have provided a new understanding of the endometrial structure and provided tractable models for the study of endometriosis. The development of organoids using cells from human patients has facilitated research into in vivo mechanisms [331]. In addition, 3D models focused on the study of endometriosis have enabled comprehensive investigations into the development and pathogenesis of this disease. Some studies using organoids derived from the ectopic and eutopic endometrium from endometriosis patients have been able to show the epigenetic mechanisms that underlie this disease [332]. Other studies have used organoids as an experimental model to study the methylation of genes and their cofactors, such as HOX genes [332,333], or progesterone signaling [334] in endometriosis. Some investigators have developed organoid models from patients with endometriosis to show the diversity within this disease, as well as providing a promising tool for drug screening [335]. There are many benefits to recreating a 3D in vivo environment, but it needs to be kept in mind that endometriosis is a multicellular disease that involves interaction between epithelial cells, stromal fibroblast, immune cells, and the extracellular matrix. Some studies using spheroids to study endometriosis have shown that the inclusion of epithelial and stromal cells with peritoneal cells could mimic the early events of the disease in vitro. These emerging methods that permit the recapitulation of several characteristics of endometriotic lesions will expand the ability to dissect the mechanisms that can contribute to endometriotic lesion establishment [336]. Furthermore, 3D models can be extremely important for drug screening, which could contribute to the development of personalized treatments for endometriosis patients.

The application of new methodologies and major advances in sequencing and analytical methods to investigate endometriosis have important implications for finding new targets and non-invasive diagnostic methods for this disease. These powerful tools will no doubt provide new avenues for the treatment of endometriosis.

## Figures and Tables

**Figure 1 ijms-25-05815-f001:**
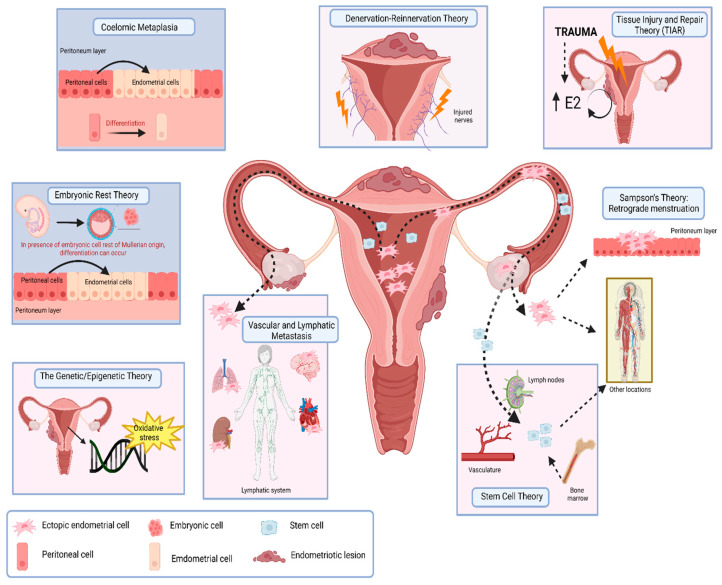
Different theories on the etiology of endometriosis. Created with BioRender.com.

**Figure 2 ijms-25-05815-f002:**
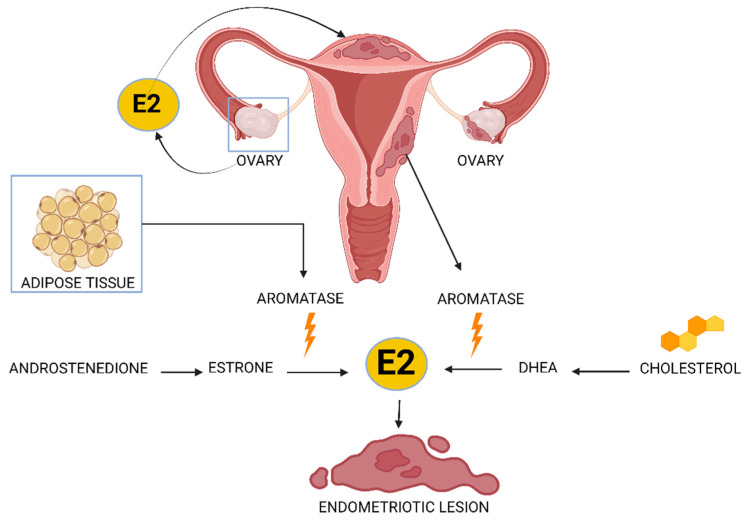
The three major sources of estradiol (E2) within endometriosis. Created with BioRender.com.

**Figure 3 ijms-25-05815-f003:**
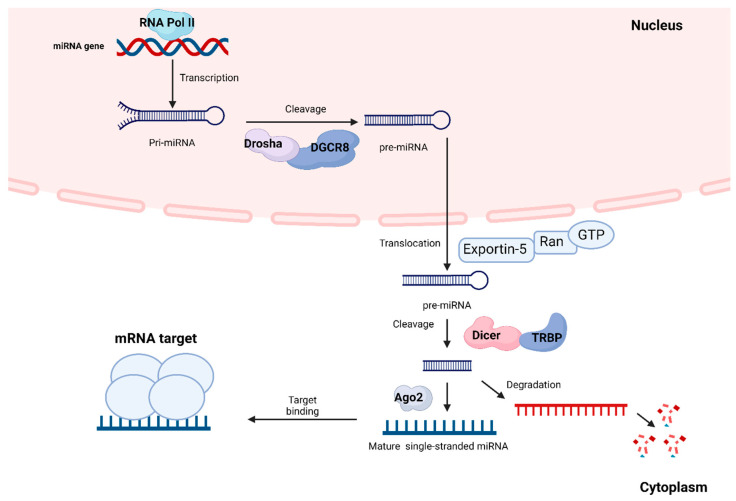
Canonical miRNA biogenesis pathway. Created with BioRender.com.

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
