# Peer review of "The Known, the Unknown and the Future of the Pathophysiology of Endometriosis"

_ijms, 2024, doi:10.3390/ijms25115815_

Round 1

Reviewer 1 Report

Comments and Suggestions for Authors

The main focus of this non systematic review is endometriosis pathophysiology, with a particular interest on latest pathogenetic mechanisms (e.g. microRNAs and epigenetic theory) and on scientific research methods.

Classical theories are mentioned, meanwhile novel evidence about inflammatory mechanisms is widely explored. A detailed analysis of cytokines pathways and fibrotic evolution of the disease is provided. The authors analyse infertility linked to endometriosis, based on the pathogenetic aspects mentioned above. An overview about microRNAs’ role in endometriosis development and diagnosis is given. Endometriosis management and treatment of endometriosis are briefly discussed.

A large section of the review investigates the animal models used to study the disease, with a focus on baboon and mouse models. Induction methods for the development of endometriosis lesions are analysed.

COMMENTS:

1. This is a non-systematic literature review. Inclusion and exclusion criteria for the selection of the articles should be provided. The time interval of the publications included should be defined. 

2. The paper appears to long with statements too often repeated in the different chapters; an accurate review is essential. The " Animal models to study the disease" chapter should be summarized.

3. Lines 47-48

“The gold standard for the diagnosis of endometriosis is surgical assessment by laparoscopic visualization.”

According to the latest ESHRE Guidelines (2022) “Laparoscopy is no longer the diagnostic gold standard and it is now only recommended in patients with negative imaging results and/or where empirical treatment was unsuccessful or inappropriate.”

4. Lines 349-351

“In cases where women with endometriosis also suffer from infertility, or present with recurrent pregnancy loss, the uNK cytotoxic activity is even higher.”

In the previous lines is written that uNK cytotoxic activity is low in the normal endometrium and it is even lower in women with endometriosis, then it can be read that the uNK cytotoxic activity is higher in women with endometriosis and infertility or recurrent pregnancy loss. This concept requires a better explanation.

5. Lines 534-540

“[…]TFG-β, is growth factor that is crucial for female reproduction. A dysregulation of the TFG-β signaling pathway may have catastrophic consequences, leading to reproductive diseases, disturbing embryo implantation, ovulation or decidualization. There is evidence that TFG-β may play a role in the etiology of endometriosis. TFG-β is also known as a mediator of P4 action during the secretory phase and modulates expression of critical proteins, including leukemia inhibitor factor (LIF), implying a mechanism by which fertility might be compromised”

More information is needed for a better comprehension of this issue.

6. Lines 603-604

“MiR-143 is highly expressed during endometriosis, inhibiting cell proliferation.”

The dysregulation of cell proliferation in a proliferative sense is mentioned among the pathophysiological mechanisms of endometriosis. Better explain why miR-143 is highly expressed in endometriosis and has an anti-proliferative function.

Reviewer 2 Report

Comments and Suggestions for Authors

What a great summary of endometriosis. I really appreciate the content and the fact that it is up to date. The team I work with have established endometriosis organoids, so I really liked that you ended the article with that. 

In line 34 you write: Endometriosis is usually classified according to the criteria formulated by the American Society of Reproductive Medicine (ASRM). 

As I am from Germany and here in Central Europe we use the #ENZIAN score to classify endometriosis, I think it is worth mentioning.  Not because I want to actively promote it. I don't have any conflict of interest with ENZIAN. But the old r-asrm classification system does not really help the patient or the clinician. With ENZIAN we have a tool to classify endometriosis at the time of diagnosis by ultrasound or MRI and at the time of surgery... 

I think it would be worth writing a paragraph about the problem of r-arsm and the ongoing developments aagl/ENZIAN.... not in the introduction but in the subtitle: 5. Management and treatment in current clinical practice.

On line 43 you write: Some of the symptoms associated with endometriosis are painful menstruation (dysmenorrhoea), cyclical or non-cyclical abdominal pain, recurrent painful urination (dysuria), pain during and after sexual intercourse (dyspareunia), painful bowel movements (dyschezia), gastrointestinal discomfort and decreased libido[11]. 

The quote from Vercellini (11) does not mention the symptom of decreased libido. There are papers in pubmed that use questions about female sexual function and conclude that decreased libido is a symptom of endometriosis. Please include these.
